# The PD-L1 Expression and Tumor-Infiltrating Immune Cells Predict an Unfavorable Prognosis in Pancreatic Ductal Adenocarcinoma and Adenosquamous Carcinoma

**DOI:** 10.3390/jcm12041398

**Published:** 2023-02-09

**Authors:** Zhiwei Zhang, Qunli Xiong, Yongfeng Xu, Xuebin Cai, Lisha Zhang, Qing Zhu

**Affiliations:** Abdominal Oncology Ward, Cancer Center, West China Hospital of Sichuan University, Chengdu 610041, China

**Keywords:** tumor-infiltrating immune cells, tumor microenvironment, immunoscore, prognosis, PD-L1, adenosquamous carcinoma of the pancreas

## Abstract

The tumor microenvironment (TME) plays a vital role in the development, progression, and metastasis of pancreatic cancer (PC). The composition of the TME and its potential prognostic value remains to be fully understood, especially in adenosquamous carcinoma of pancreas (ASCP) patients. Immunohistochemistry was used to explore the clinical significance of CD3, CD4, CD8, FoxP3, and PD-L1 expression within the TME and to identify correlations with the prognosis of PC in a series of 29 patients with ASCP and 54 patients with pancreatic ductal adenocarcinoma (PDAC). Data from the Gene Expression Omnibus (GEO) and the Cancer Genome Atlas (TCGA) were accessed to obtain the scRNA-seq data and transcriptome profiles. Seurat was used to process the scRNA-seq data, and CellChat was used to analyze cell–cell communication. CIBERSORT was used to approximate the constitution of tumor-infiltrating immune cell (TICs) profiles. Higher levels of PD-L1 were linked with a shorter overall survival in ASCP (*p* = 0.0007) and PDAC (*p* = 0.0594). A higher expression of CD3+ and CD8+ T-cell infiltration was significantly correlated with a better prognosis in PC. By influencing the composition of tumor-infiltrating immune cells (TICs), high levels of PD-L1 expression are linked with a shorter overall survival in ASCP and PDAC.

## 1. Introduction

Pancreatic cancer (PC) is the sixth most common cause of cancer deaths worldwide and is an aggressive malignancy with a poor 5-year survival rate of 11% [1]. Pancreatic ductal adenocarcinoma (PDAC) is the most common form of PC that accounts for 85% of all cases [2]. Adenosquamous carcinoma of the pancreas (ASCP) is an uncommon subtype of PC that has an estimated survival of 0.38 to 10% [3] and has significantly worse clinical outcomes compared to PDAC [4,5].

Generally, PC has an immunosuppressive microenvironment with rare antitumor T-cell infiltration [6]. Immunotherapy drugs such as immune checkpoint inhibitors are only effective in a small percentage of PC patients, highlighting the importance of clarifying the reason for the limited efficacy of checkpoint blockade [7,8,9]. PC cells can evade several types of antitumor immune cells through particular tumor-infiltrating immune cells (TICs), myeloid-derived suppressor cells, and M2 macrophages [7]**,** demonstrating that TICs are key factors in driving responses to therapy and classifications of PC [10].

A growing body of research suggests the tumor microenvironment (TME), particularly the accumulation of TICs, affects the prognosis of PC patients [11,12,13]. The quantitative assessment of TICs can be a more precise predictor of survival in colorectal cancer compared to the classical TNM classification system [14,15]. A study reported that PC patients with high levels of CD3+, CD4+, and CD8+ T cells have significantly prolonged survival [16]. Additionally, it has been shown that Foxp3+ Tregs expression is correlated with a poor prognosis in PC due to its antitumor immunity-suppressing effects [17]. Previous studies have demonstrated that the expression level of PD-L1 is associated with differential prognostic effects in various cancers [18,19]. Compared to traditional TNM staging, the quantification of TICs within the TME can be used to more accurately predict prognosis in PC [20,21]. However, a complete understanding of the underlying biology and roles of the TME in PC patients remains challenging, particularly in ASCP due to its very low incidence. Hence, genetic- and tissue-level analyses are required to better understand the dynamic transition of the TME and to delineate the mechanisms underlying the development and progression of PC.

This study aimed to investigate differences in the composition of immune cells within the TME in PDAC and ASCP. The potential impact of TICs on overall survival was investigated in a cohort of PC patients, analyses were conducted based on the ImmuneScore and single-cell RNA sequencing (scRNA-seq) data.

## 2. Materials and Methods

### 2.1. Patients and Involved Samples

This study included paraffin specimens from 29 ASCP patients who underwent a pancreaticoduodenectomy from 2014 to 2019 at the West China Hospital, Sichuan University. As a control, tumor specimens from 54 PDAC patients matched with ASCP cases for sex and age were also chosen. The diagnostic pathology report for specimens of ASCP and PDAC was provided by the Pathology Department, West China Hospital of Sichuan University. Patients with PC who had received neoadjuvant chemoradiotherapy, had other cancers, or were lost during follow-up were excluded from the analysis. PC patients were retrieved from our database to acquire detailed information on treatments, pathological tumor characteristics, surgical treatments, and follow-up. The data were further confirmed by checking against the patients’ medical records. The patient information is shown in Table 1.

Transcriptome profiles of 183 PDAC cases were obtained from The Cancer Genome Atlas (TCGA) [22]. The clinical data of PDAC patients are presented in Appendix A. The Gene Expression Omnibus (GEO) is a public database that provides functional genomic information [23]. The scRNA-seq data of PDAC and ASCP were obtained from the GEO database, including PDAC samples from GSE111672 [24] and one ASPC sample from GSE165399 [25].

### 2.2. Immunohistochemistry (IHC)

Representative tumor tissues were cut into 3 μm thick tissue sections and used for immunohistochemical staining with anti-human CD3 antibody (Proteintech, 17617-1-AP, 1:1000), anti-human CD4 antibody (HUABIO, ST0488, 1:500), anti-human CD8 antibody (Proteintech, 66868-1-1g, 1:5000), anti-human FoxP3 antibody (Abcam, ab20034, 1:500), and anti-human PD-L1 antibody (Proteintech, 66248-1-1g, 1:10,000). Tissue sections were immersed in tris-EDTA buffer (pH 9.0) and heated in a microwave for 16 min. The sections were incubated with the diluted primary antibodies overnight in a refrigerator at 4 °C, followed by incubation with the corresponding secondary antibodies for 2 h at room temperature. We used DAB as the chromogen and hematoxylin as the counterstain.

### 2.3. Immunohistochemical Score

We assessed the immunohistochemical staining scores without knowledge of the clinical data. The stained sections were scanned using a 3D-Histech Pannoramic-250 Flash II slide scanner (3D-Histech, Budapest, Hungary), and the images were analyzed using ImageJ. First, we searched for five hotspots with higher infiltration in each tissue section and calculated the immune score for the hotspots. Then, the final score was made as the average of the five scores. We quantified PD-L1 expression using the IHC Profiler ImageJ Plugin [26], which is a program for ImageJ that automatically assesses the intensity of staining in tissue sections. We chose the intratumoral area with a high expression of PD-L1, and the formula calculated the IHC score of these areas: (% area of weak staining) + (2 × % area of moderate staining) + (3 × % area of strong staining). Then, we received a score between 0 and 300. The IHC scores less than 75 were defined as the low-expression group, and scores between 75 and 300 were delivered into the high-expression group. Other tissue sections were identified using a previously reported method [27]. The separation of the color layers was based on hematoxylin and diaminobenzidine. An image analysis was then performed using luminance thresholding with ImageJ. The selected areas with high lymphocyte counts were within the tumors and at the infiltrating edge (Figure 1A–C). The positive cells were measured with ImageJ. The CD3, CD4, CD8, and FoxP3 samples were divided into two groups based on counts of the positively stained lymphocytes per unit area, defined as high and low cell densities (Figure 1D). We used Xtile (version 3.6.1, https://github.com/xryanglab/xtail, accessed on 8 May 2022) to determine the cutoff values (Appendix A) that had the highest sensitivity and specificity. The number and percentage of patients in each group are shown in Appendix A.

### 2.4. Integration, Clustering and Cell Type Identification

Quality control for single-cell samples was based on the number of detected genes and the percentage of mitochondrial, ribosomal and hemoglobin genes. After that, Seurat (version 4.2.0) was used to process the data [28]. Firstly, Seurat was developed for integrating scRNA-seq. Then, cell type identification and clustering were performed. Next, we chose the FindClusters parameter resolution of 0.8 to achieve cell clustering. Finally, CellChat (version 1.5.0) was utilized for the cell–cell communication analysis.

### 2.5. Assessment of Tumor-Infiltrating Immune Cells (TICs)

To investigate the immune-associated TME, CIBERSORT was used to assess the proportion of TICs in PDAC cases. CIBERSORT is a powerful and novel method to characterize the cellular composition of solid tumor tissues based on gene expression profiles. It outperforms other methods when dealing with unknown contents, closely related cell types, and background noise [29].

### 2.6. Assessment of TME Parameters (ImmuneScore, StromalScore, and ESTIMATEScore)

The ESTIMATE package in R (software version: 4.1.1) was used to calculate the ImmuneScore (Immune Component Ratio), StromalScore (Matrix Component Ratio), and ESTIMATEScore (Sum of ImmuneScore and StromalScore) for each PC sample [30]. For these measures, higher scores represent higher amounts of the corresponding components in the TME.

### 2.7. Identification of Differentially Expressed Genes (DEGs)

To further analyze differences in the different immune scores at the gene level, DEGs were identified by the differential analysis of high- and low-immunity cases using the “limma” R package with a false discovery rate (FDR) <0.05 and an absolute value of log2-fold change (FC) >1. To explore the biological functions and signaling pathways of the DEGs, we used the R packages “clusterProfiler”, “enrichplot”, and “ggplot2” for the GO and KEGG enrichment analyses. The R package “pheatmap” was used to draw heatmaps of the DEGs.

### 2.8. Statistical Analysis

SPSS Statistics 23.0 (IBM, Armonk, NY, USA) was used for statistical analysis. The Kaplan–Meier method was used to estimate the survival probability. The survival time was calculated from the date of surgery to the time of death or follow-up (31 September 2021). Cox proportional hazards were applied for univariate and multivariate analyses to assess the important factors. A *p*-value threshold of <0.05 was used to indicate statistically significant differences.

## 3. Results

### 3.1. High PD-L1 Expression Is Associated with Shorter Overall Survival

By IHC staining in our data to detect the levels of PD-L1 protein, we found that PD-L1 expression was linked with survival outcomes. Representative images of positive PD-L1 immunostaining are presented in Appendix A. A higher level of PD-L1 expression was linked with shorter OS in ASCP (*p* = 0.0007; Figure 2A) and PDAC (*p* = 0.0594; Figure 2B) and PC (*p* = 0.0089; Figure 2C). To validate the inference, the RNA expression of PD-L1 was found to associate with the clinicopathological characteristics and survival outcomes of PDAC patients from TCGA data, which are shown in Appendix A.

### 3.2. CD3+, CD4+, CD8+, and FoxP3+ T Cells Infiltrate Pancreatic Tumors and Correlate with Survival

To demonstrate that the composition of TICs in PC was correlated with survival, IHC staining was performed to validate antibodies for detecting various immune biomarkers (FoxP3+, CD3+, CD4+, and CD8+ T cells) in PC samples. We then assessed the expression of these markers in 29 ASCP and 54 PDAC samples. Representative images (Appendix A) have shown high and low levels of CD3+, CD4+, CD8+, and FoxP3+ T-cell infiltration.

The Kaplan–Meier survival analysis showed that lower levels of CD3+ T cells were related to poorer overall survival rates in ASCP (*p* = 0.0353; Figure 2D), PDAC (*p* = 0.0034; Figure 2E), and PC (*p* = 0.0211; Figure 2F). Similarly, higher levels of infiltrating CD8+ T cells were related to longer OS in ASCP (*p* = 0.0841; Figure 2J), PDAC (*p* = 0.0277; Figure 2K), and PC (*p* = 0.0298; Figure 2L). However, CD4+ (Figure 2G–I) and FoxP3+ (Figure 2M–O) T cells were shown no significant association with survival. Several parameters were used in the univariate analysis of the Cox proportional hazards model (age, sex, histological grade, T stage, N stage, M stage, TNM stage, perineural invasion, hypertension, diabetes, smoking, drinking, recurrent status, CD3, CD4, CD8, FoxP3, and PD-L1 expression; Table 2). Higher T and N stages were considered to be protective factors, yet these results may be potentially confounded by the small number of ASCP samples. The multivariate analysis demonstrated that PD-L1 expression was closely related to overall survival in ASPC (Table 3).

### 3.3. Cell Clustering of PDAC and ASCP

After quality control, 8327 cells were obtained for subsequent analysis, and these cells were sorted for the cell cycle score. Next, 0.8 was set to reveal the subgroup (Appendix A). The cells were cataloged into subgroups annotated with marker genes acquired from CellMarker 2.0 [31] (Figure 3D). As a result, T cells, B cells, macrophages, cancer cells, acinar cells, ductal cells, endothelial cells, and epithelia cells were identified (Figure 3A). The distribution and percentage of PDAC and ASCP are presented in Figure 3B,C. The highly expressed genes in each type of cells are shown in Figure 4A. Summarily, ductal cells and epithelial cells mainly existed in the PDAC and were present at low levels in the ASCP samples. In our study, the proportion of T cells (3.50%) and B cells (2.71%) was rare in PDAC. However, T cells (9.26%) and B cells (23.73%) were highly infiltrated in ASPC.

### 3.4. Dynamic Interaction between Tumor Cells and Immune Cells

We used CellChat to explore putative cellular interactions and communication (receptor–ligand pairs) between cancer cells and macrophages, T cells, and B cells in PC patients evaluated by scRNA-Seq. Laminin, MHC-I, APP, MK, and annexin are significantly involved in cellular interaction and communication for cancer cells and immune cells (Figure 4B). Tumor cells can directly or indirectly affect immune cells through various signals, such as the annexin pathway (Appendix A). The dynamic interaction between cancer cells and immune cells has a vital impact on the prognosis of PC patients. The expression levels of CD3, CD4, CD8, FoxP3, and PD-L1 in each cell type are shown in Figure 4C. By detecting the expression of immune-related markers, we can effectively predict the prognosis of PC patients.

### 3.5. The Proportion of TICs Was Rare in PC

CIBERSORT was applied to construct 22 subtypes of immune cells and to analyze the proportion of TICs in the PDAC cases (Figure 5A,B). The five most prevalent immune cell types in PDAC included M2 macrophages, resting CD4 memory T cells, M0 macrophages, CD8 T cells, and naive B cells. The 22 subtypes of TICs in PC were correlated with each other (Figure 5C). Macrophages and resting CD4 memory T cells may be linked to an immunosuppressive microenvironment in PC.

### 3.6. DEGs Assessed by the ImmuneScore

After generating the ESTIMATE scores (ImmuneScore, StromalScore, and ESTIMATEScore), we found that the ImmuneScore plays an essential role in the progression of PDAC (Appendix A). We divided PC cases into high- and low-immunity groups by the median ImmuneScore for a further comparative analysis, and 826 DEGs were totally identified, with a downregulation of 9 genes and an upregulation of 817 genes in PDAC (Figure 5D).

The top 20 downregulated and upregulated genes in PDAC were assessed by the absolute value of log_2_ FC. The heatmaps are shown in Figure 5E. In PDAC, CCL17 (C-C Motif Chemokine Ligand 17) and IGHM (Immunoglobulin Heavy Constant Mu) were highly expressed in the high-immunity group. The GO analysis indicated that the DEGs were chiefly correlated with immune functions such as leukocyte-mediated immunity and the upregulation of leukocyte activation in PDAC (Figure 5F). Likewise, the KEGG enrichment analysis showed high enrichment for immune biological processes such as hematopoietic cell lineages and cytokine–cytokine receptor interactions (Figure 5G). Consequently, the main functions of the DEGs were often immune-related biological processes, proving that immune components are important parts of the TME in PDAC.

### 3.7. PD-L1 Expression Is Related to the Proportion of TICs

The results from the Wilcoxon rank sum test showed that the level of PD-L1 expression in the normal samples was significantly lower than the tumor samples (Figure 6A). For the 22 subtypes of infiltrating immune cells in PDAC, two TICs were positively associated with the levels of PD-L1 expression, including memory-activated T cells CD4 and M1 macrophages. Three types of TICs were negatively associated with PD-L1 expression, including regulatory T cells (Tregs), activated NK cells, and activated mast cells (Figure 6B,C). These data suggested that PD-L1 expression could significantly influence the proportion of TICs in the TME (Figure 6D).

## 4. Discussion

In this study, we showed that the composition of TICs can further our understanding of prognosis and treatment in pancreatic cancer. Although recent studies have focused on immune cell compositions in the PDAC microenvironment and its prognostic value [16,32,33,34], none of these studies have discussed the predictive value and constitution of immune cells in ASCP. What’s more, we chose PD-L1, which was rarely detected in previous studies, as an immune marker. Additionally, we have analyzed the scRNA-seq data and transcriptome profiles to explain the prognostic value of TICs in pancreatic cancer. This was not seen in prior studies.

In our study, immunohistochemical staining was used to demonstrate that higher levels of CD3+ and CD8+ T cell infiltration were linked with a better prognosis in PC. Therefore, CD3 and CD8 can be considered prognostic biomarkers in PC, but CD4 was not. CD3 is part of the T-cell receptor/CD3 complex and is involved in T-cell development and signal transduction. The expression of CD3 is typically associated with T cells infiltrating. CD8+ cytotoxic T cells (CTL) play a vital role in recognizing and killing cancer cells. However, CD4+ cells play a complex role in tumorigenesis and tumor progression. On the one hand, CD4+Th cells and CD4+ CTL can enhance the antitumor immune response. On the other hand, CD4+ Tregs can suppress antitumor immunity. During the immunohistochemical, we found the infiltration of FoxP3+ T cells is at a low level in pancreatic cancer. The difference in expression level was modest between groups with high and low FoxP3. In our study, increased numbers of FoxP3+ lymphocytes are no significant correlations with poor prognosis. More cases are needed to confirm this.

By analyzing the RNA-seq data from TCGA and scRNA-seq data from GEO, we have found more details about TICs in PC. Analysis of the composition of the TICs showed that the five most common immune cell types were M2 macrophages, resting CD4 memory T cells, M0 macrophages, CD8 T cells, and naive B cells, which would be seen as immunosuppressive TME [32]. An immunosuppressive TME and low immunogenicity may account for the poor efficacy of immunotherapy in PC [33]. Dynamic interactions between cancer cells and immune cells can prevent the expansion of cytotoxic T cells and promote the production of myeloid-derived suppressor cells (MDSCs), leading to tumor growth and invasion [33,34]. The PD1/PD-L1 pathway and Treg cells are both known to contribute to tumor immune tolerance. Remarkably, we found high expression of PD-L1 was associated with low infiltration of Treg cells. We considered that the PD-1/PD-L1 axis could inhibit Ras/MEK/ERK pathway to repress T cell proliferation [35], including Treg cells. In addition, the PD-1/PD-L1 axis leads to a decrease in cell apoptosis-related gene Bcl-xl and promotes Treg cell apoptosis [36]. The high level of PD-1/PD-L1 expression resulted in the apoptosis of Treg cells and the decrease of proliferation, leading to the reduction of Treg cells in the TME. Although the microenvironment was both immunosuppressed in PDAC and ASCP, a higher proportion of immunosuppressive cells were found in ASCP compared to PDAC [25]. These observed differences in the immune TME may account for the faster progression and worse prognosis of ASCP compared to PDAC. Regrettably, our study does not prove this point, which needs to be elucidated by future studies.

Therefore, measures that can increase T cell infiltration in TME can make a better prognosis for patients with ASCP. The blockade of the PD-1/PD-L1 pathway, regulating gut microbiota, CAR-T cell therapy, and immunogenic cell death (ICD) can be considered to increase T cell infiltration in TME.

We demonstrated that PD-L1 was overexpressed in PC. Moreover, the expression of PD-L1 can influence immune cells within the TME in ASCP and PDAC suggesting the differential efficacy of PD-L1 inhibitors in PC. High PD-L1 expression was linked with an increased number of macrophages in the TME, and tumor-associated macrophage infiltration was found to be correlated with poor prognosis in most tumors [37]. A previous study showed that PD-L1 directly targets FOXP3 to promote immune evasion by recruiting Treg cells in PDAC [38].

The blockade of the PD-1/PD-L1 pathway could reactivate antitumor immunity and raise the numbers of infiltrating T cells [39,40]. PD-L1 disables antitumor immune responses by binding to PD-1 [41], and regulates the development of Treg cells by modulating the expression of key signaling molecules such as the downregulation of Akt and mTOR, and the upregulation of PTEN. A study showed that PD-L1 blocks infiltrating CD8+ T cells and anti-PD-L1 mAbs can induce significant antitumor effects in mouse models of PC [42]. Additionally, PD-1/PD-L1 blockade on the group 2 innate lymphoid cells (ILC2s) can expand ILC2s and increase T cell infiltrating in PC [43]. These findings suggest that PD-L1 can affect the composition of TICs in the TME through multiple pathways. Immunotherapy targeting PD-L1 immune checkpoints has demonstrated clinical benefits in some solid tumors but has limited efficacy in PC patients. Immune checkpoint inhibitors require specific T cells to kill tumor cells that also require physical contact with cancer cells [44]. Further studies are needed to increase the efficacy of PD-L1 immunotherapy in PC patients.

Previous trials have suggested that the gut microbiota can metabolize trimethylamine N-oxide (TMAO) to increase the number of antitumor TICs in PC, such as CD8+ T cells [45]. These findings suggest that gut microbes can improve the prognosis of PDAC patients by increasing the immune cell infiltration within the TME [46]. Patients with effective PD-1/PD-L1 blockade often have colonies of Bifidobacterium longum, Collinsella, and Enterococcus faecalis. When the intestinal flora of responding epithelial cancer patients is introduced into germ-free mice, the number of infiltrating T cells increases to improve the efficacy of anti-PD-L1 therapy [47,48]. In gastrointestinal tumors, the gut bacteria are capable of producing single-chain fatty acids that are positively associated with responses to anti-PD-1/PD-L1 including eubacteria, lactobacilli, and streptococci [49]. Therefore, gut microbes have major potential in the treatment of PC.

Chimeric antigen receptor T (CAR-T) cell therapy is a new field of immune engineering and belongs to adoptive cell therapy (ACT). T cells are genetically modified in vitro and injected into the body to produce antitumor or immunomodulatory effects. CAR helps T cells recognize, activate, proliferate, and increase T cell infiltration in TME [50]. As a new method of immunotherapy, immunogenic cell death (ICD) uses inactive cancer cells as cell-based vaccines [51]. The immunogenicity of ICD is mainly mediated by DAMP and cytokines/chemokines, which are recognized by the corresponding pattern recognition receptors, leading to the activation of the immune systems [52]. It has been demonstrated that CAR-T cell therapy and immune therapy based on ICD could increase immune infiltration, which has great prospects in the treatment of pancreatic cancer.

Despite the compelling observations presented in this report, our study had several limitations. We only analyzed the gene expression profiles of one ASCP case from the GEO database, as limited sequencing information was available. Similarly, the ASCP sample size for IHC analysis was also small which may confound our findings. Then, we used Xtile to find cut-off points which could make a statistically significant difference in the survival curves. More ASCP patient data were needed to validate our results.

## 5. Conclusions

In conclusion, PDAC and ASCP have immunosuppressive microenvironments that contain different proportions of TICs. The DEGs obtained from ImmuneScore, GO, and KEGG enrichment analysis highlighted differences between PDAC and ASCP. PD-L1 expression was correlated with the constitution of TICs and was associated with OS in PC. Our findings stress the importance of TICs to accurately predict the prognosis and the efficacy of PD-L1 inhibitors in the treatment of PC.

## Figures and Tables

**Figure 1 jcm-12-01398-f001:**
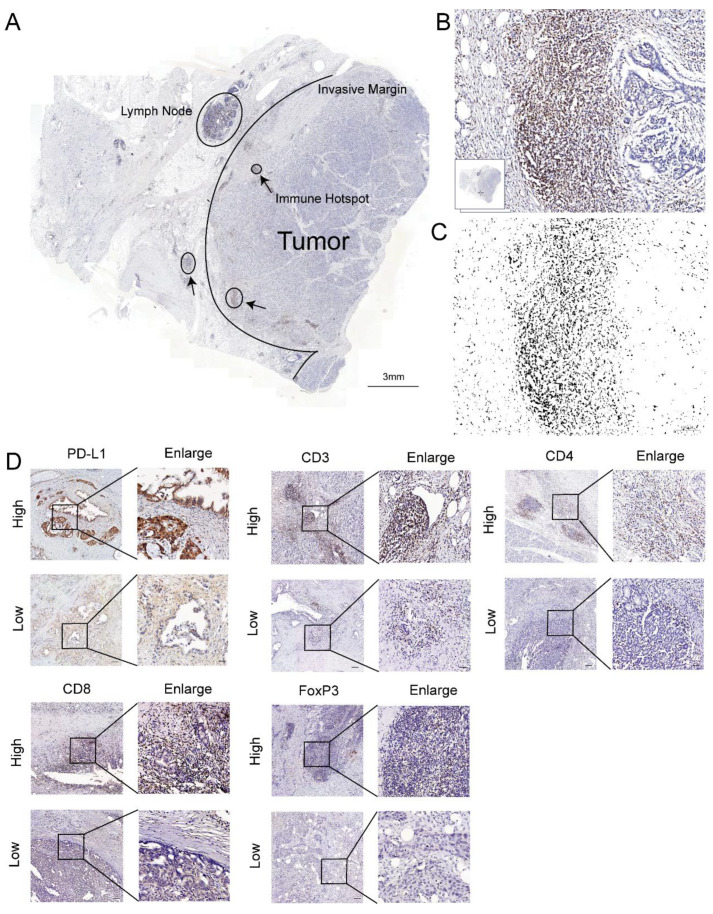
T-cell infiltration marker and PD-L1 in pancreatic cancer. (**A**) Image of a whole section of anti-CD3-stained tissue displaying the regions of tumor and the invasive margin. The immune hotspots are shown with arrows. (**B**) A hotspot in the tumor shows numerous positive T cells and fewer tumor cells. (**C**) The corresponding image analysis shows the counted cells. (**D**) Representative pictures for low and high CD3, CD4, CD8, FoxP3, and PD-L1 immunostaining (brown) in PC tumors. Original magnification, 20×; scale bars, 100 µm. Enlarged images on the right panel are shown for the areas outlined by black squares. Scale bars show 25 µm in length.

**Figure 2 jcm-12-01398-f002:**
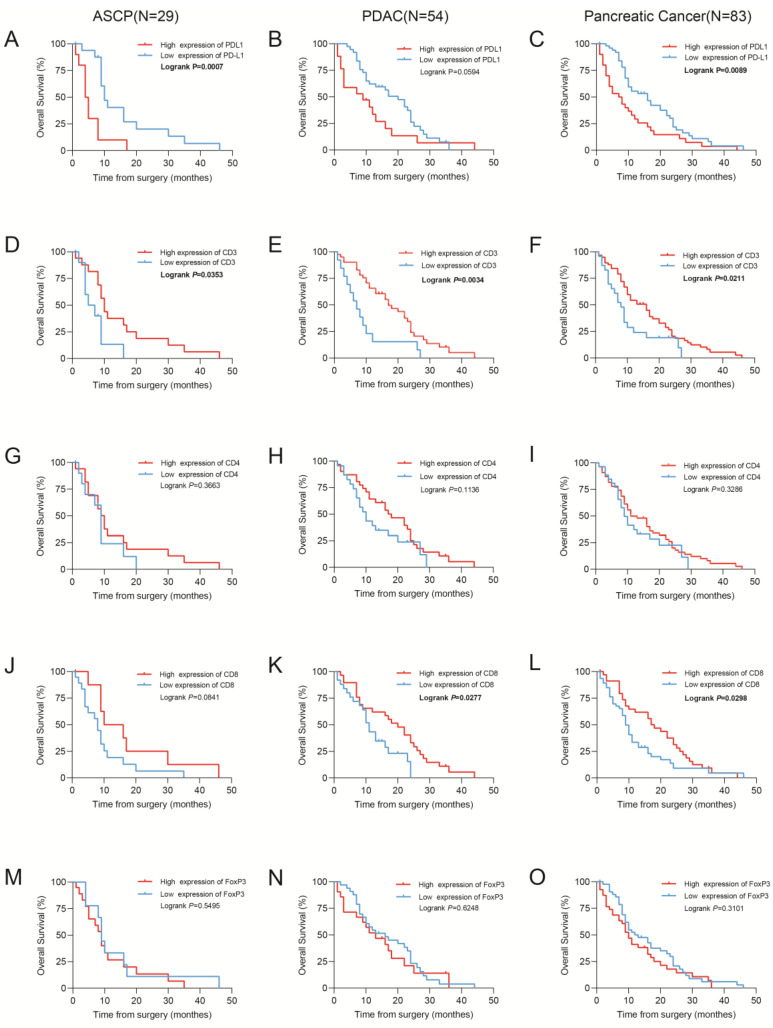
Kaplan–Meier estimates of overall survival. (**A**–**E**) Kaplan–Meier curves for overall survival according to the PD-L1, CD3, CD4, CD8, and FoxP3 immunoscores in cohorts of patients with adenosquamous carcinoma of the pancreas (ASCP). (**F**–**J**) Kaplan–Meier curves for overall survival according to the PD-L1, CD3, CD4, CD8, and FoxP3 immunoscores in cohorts of patients with pancreatic adenocarcinoma (PDAC). (**K**–**O**) Kaplan–Meier curves for overall survival according to the PD-L1, CD3, CD4, CD8, and FoxP3 immunoscores in cohorts of patients with pancreatic cancer. A total of 83 patients (29 ASCP patients and 54 PDAC patients) were included in the pancreatic cancer group.

**Figure 3 jcm-12-01398-f003:**
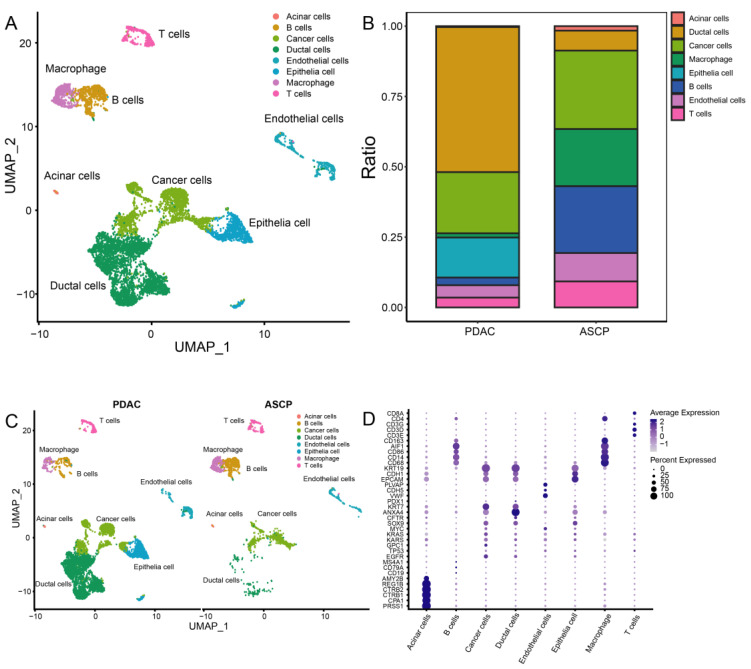
A total of 8327 single cells were clustered into ten subgroups based on scRNA-seq. (**A**) UMAP plot showing eight cell types in tissues of the PC. (**B**) Percentage of cell types between PDAC and ASCP. (**C**) Cell distribution in the PDAC and ASCP. (**D**) Annotation of each subgroup with markers.

**Figure 4 jcm-12-01398-f004:**
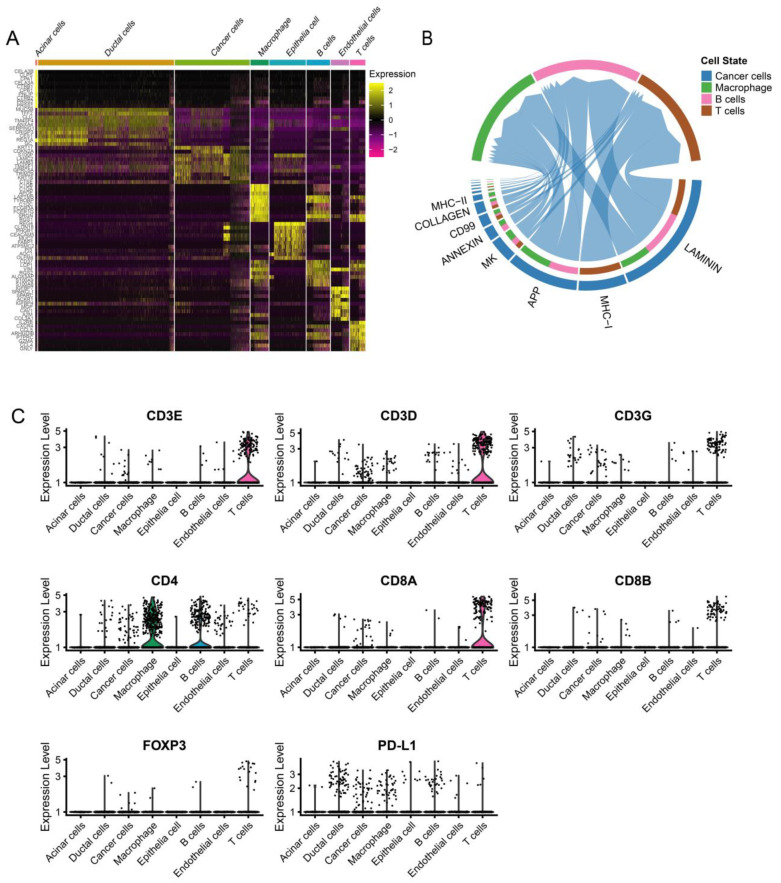
ScRNA-seq demonstrating specific cell subpopulations and cell communications among subgroups. (**A**) Heatmap of top ten differentially expressed genes in each cell type. (**B**) CellChat has been used to quantitively infer and analyze intercellular communication networks from scRNA-seq data. (**C**) Violin plot showing the expression of immunomarkers of eight cell clusters.

**Figure 5 jcm-12-01398-f005:**
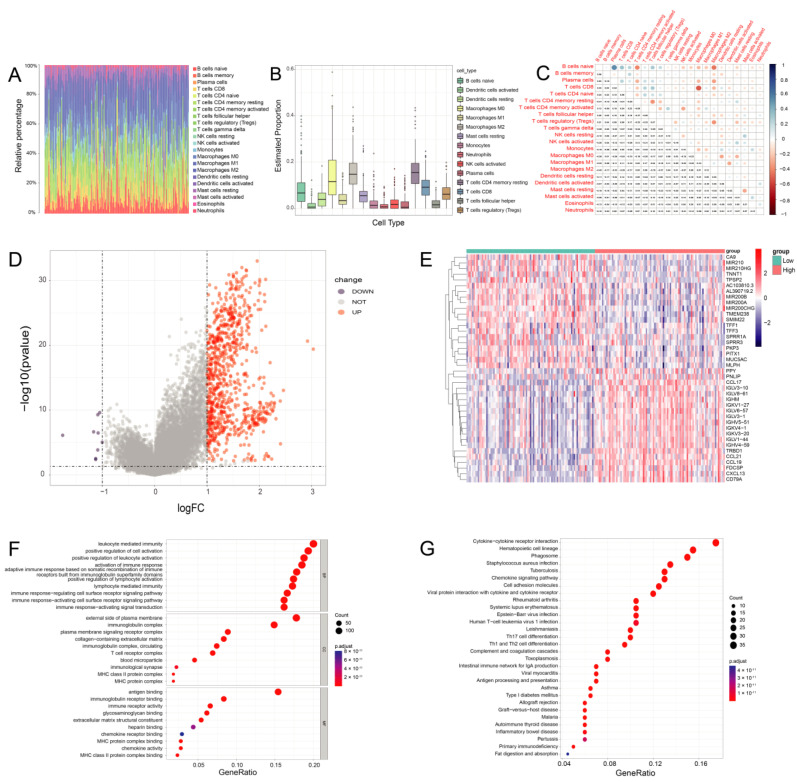
The immune cells composition of PDAC. (**A**) Bar plot shows the proportion of 22 types of TICs in PDAC tumor samples. The column names of the plot were sample IDs. (**B**) The estimated proportion of 22 types of TICs in PDAC tumor samples. (**C**) Heatmap shows the correlation between 22 kinds of TICs and numerics in each tiny box, indicating the *p*-value of the correlation between two cells in PDAC. (**D**) Volcano plot for DEGs. The blue and red dots represented the significantly downregulated and upregulated genes, respectively, and the gray dots represented the genes without differential expression. (**E**) Heatmap for DEGs generated by comparison of the high-score group vs. the low-score group in ImmuneScore. The row name of the heatmap is the gene name, and the column name is the IDs of samples not shown in the plot. DEGs were determined by the Wilcoxon rank sum test, with FDR < 0.05 and |log2 FC | > 1 as the significance threshold. (**F**,**G**) GO and KEGG enrichment analyses for DEGs in PDAC, and terms with *p* and q < 0.05 were believed to be enriched significantly.

**Figure 6 jcm-12-01398-f006:**
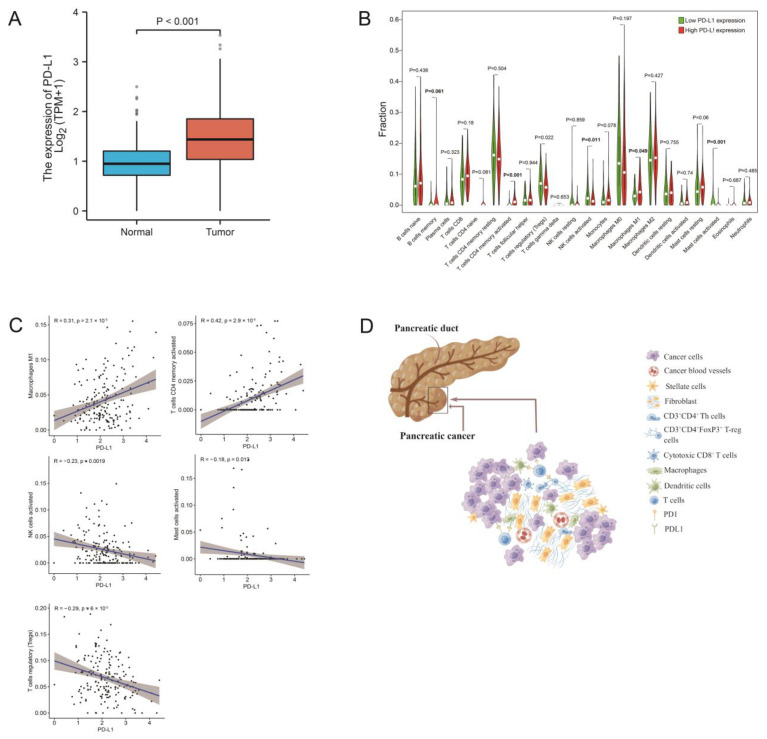
The correlation of TIC proportions and PD-L1 expression in pancreatic cancers. (**A**) Differentiated expression of PD-L1 in the normal and tumor samples. Analyses were conducted across all normal and tumor samples by the Wilcoxon rank sum test. (**B**) Violin plot showed the ratio differentiation of 22 types of immune cells between PDAC samples with low or high PD-L1 expression. (**C**) The scatter plot shows the correlation of 5 kinds of TIC proportions with the PD-L1 expression in PDAC (*p* < 0.01). The Pearson coefficient was used for the correlation test. (**D**) Tumor cells affect the composition of the immune microenvironment through PD-1/PD-L1 blockade. Figure 6D was created by Figdraw.

**Table 1 jcm-12-01398-t001:** Patient demographics and clinicopathologic factors (*n* = 83).

	ASCP (*n* = 29)	PDAC (*n* = 54)	Total (*n* = 83)	*p*-Value
Age (years)	60.90 ± 9.43	60.26 ± 9.35	60.48 ± 9.38	0.77
Sex (N, %)				0.76
Male	20 (69.87)	34 (62.96)	54 (65.06)	
Female	9 (31.03)	20 (37.04)	29 (34.94)	
T Stage (N, %)				**0.04**
I	0 (0.00)	1 (1.85)	1 (1.20)	
II	4 (13.79)	2 (3.70)	6 (7.23)	
III	21 (72.41)	50 (92.59)	71 (85.54)	
IV	4 (13.79)	1 (1.85)	5 (6.02)	
N Stage (N, %)				0.54
0	23 (79.31)	47 (87.04)	70 (84.34)	
1	6 (20.69)	7 (12.96)	13 (15.66)	
M Stage (N, %)				0.91
0	27 (93.10)	52 (96.30)	79 (95.18)	
1	2 (6.90)	2 (3.70)	4 (4.82)	
TNM Stage (N, %)				0.13
I	4 (13.79)	3 (5.56)	7 (8.43)	
II	20 (68.97)	48 (88.89)	68 (81.93)	
III	3 (10.34)	1 (1.85)	4 (4.82)	
IV	2 (6.90)	2 (3.70)	4 (4.82)	
Histologic Grading (N, %)				0.14
Well differentiated	1 (3.45)	1 (1.85)	2 (2.41)	
Moderately differentiated	19 (65.52)	24 (44.44)	43 (51.81)	
Poorly differentiated	9 (31.03)	29 (53.70)	38 (45.78)	
Perineural Invasion (N, %)				0.35
Yes	15 (51.72)	35 (64.81)	50 (60.24)	
No	14 (48.28)	19 (35.19)	33 (39.76)	
Hypertension (N, %)				1.00
Yes	5 (17.24)	10 (18.52)	15 (18.07)	
No	24 (82.76)	44 (81.48)	68 (81.93)	
Diabetes (N, %)				0.64
Yes	8 (27.59)	11 (20.37)	19 (22.89)	
No	21 (72.41)	43 (79.63)	64 (77.11)	
Drinking (N, %)				**0.03**
Yes	16 (55.17)	15 (27.78)	31 (37.35)	
No	13 (44.83)	39 (72.22)	52 (62.65)	
Smoking (N, %)				0.64
Yes	9 (31.03)	21 (38.89)	30 (36.14)	
No	20 (68.97)	33 (61.11)	53 (63.86)	
Recurrent Status (N, %)				0.84
Yes	10 (34.48)	16 (29.63)	26 (31.33)	
No	19 (65.52)	38 (70.37)	57 (68.67)	

*p*-values < 0.05 were considered significant and have been bolded.

**Table 2 jcm-12-01398-t002:** Univariate analysis with Cox proportional hazard model.

	OS of PDAC (*n* = 54)		OS of ASCP (*n* = 29)	
	HR (95% CI)	*p* Value	HR (95% CI)	*p* Value
Age (years) (≥65 vs. <65)	1.20 (0.62–2.10)	0.650	1.00 (0.40–2.70)	0.960
Sex (female vs. male)	1.30 (0.73–2.50)	0.350	0.78 (0.34–1.80)	0.550
T Stage (T4+T3 vs. T2+T1)	2.50 (0.59–11.00)	0.210	0.26 (0.08–0.83)	0.023
N Stage (N1 vs. N0)	1.00 (0.43–2.50)	0.950	0.79 (0.31–2.00)	0.620
M Stage (M1 vs. M0)	16.00 (2.90–86.00)	**0.001**	1.70 (0.39–7.80)	0.470
TNM Stage (III+IV vs. I+II)	1.30 (0.39–4.30)	0.680	2.00 (0.70–5.60)	0.200
Histologic grading (poorly differentiated vs. well- and moderately differentiated)	2.40 (1.30–4.50)	**0.006**	2.00 (0.70–5.60)	0.200
Perineural invasion (yes vs. no)	1.20 (0.67–2.30)	0.490	0.61 (0.27–1.4)	0.230
Hypertension (yes vs. no)	1.70 (0.83–3.50)	0.150	0.81 (0.27–2.4)	0.700
Diabetes (yes vs. no)	1.70 (0.83–3.50)	0.150	0.52 (0.20–1.30)	0.180
Drinking (yes vs. no)	1.80 (0.93–3.30)	0.083	0.98 (0.44–2.20)	0.960
Smoking (yes vs. no)	1.40 (0.81–2.60)	0.220	0.51 (0.18–1.40)	0.190
Recurrent status (no vs. yes)	0.80 (0.43–1.50)	0.490	0.55 (0.24–1.30)	0.160
CD3 (high expression vs. low expression)	0.39 (0.20–0.76)	**0.006**	0.37 (0.15–0.92)	**0.033**
CD4 (high expression vs. low expression)	0.62 (0.34–1.10)	0.120	0.66 (0.28–1.60)	0.350
CD8 (high expression vs. low expression)	0.48 (0.25–0.93)	**0.030**	0.47 (0.19–1.20)	0.099
FoxP3 (high expression vs. low expression)	1.10 (0.58–2.00)	0.840	1.30 (0.54–3.00)	0.590
PD-L1 (high expression vs. low expression)	1.80 (1.06–4.15)	**0.047**	4.20 (1.70–10.00)	**0.002**

*p*-values < 0.05 were considered significant and have been bolded.

**Table 3 jcm-12-01398-t003:** Multivariate analysis with Cox proportional hazard model.

	OS of PDAC (*n* = 54)		OS of ASCP (*n* = 29)	
	HR (95% CI)	*p* Value	HR (95% CI)	*p* Value
Age (years) (≥65 vs. <65)	1.50 (0.65–3.30)	0.370	1 (0.21–5)	0.970
Sex (female vs. male)	1.10 (0.46–2.60)	0.830	0.34 (0.063–1.8)	0.210
T Stage (T4+T3 vs. T2+T1)	2.20 (0.28–16.00)	0.460	0.47 (0.084–2.7)	0.400
N Stage (N1 vs. N0)	3.70 (1.00–13.00)	**0.049**	0.087 (0.011–0.7)	**0.022**
TNM Stage (III+IV vs. I+II)	1.50 (0.38–6.20)	0.550	0.93 (0.22–3.8)	0.920
Histologic grading (poorly differentiated vs. well- and moderately differentiated)	2.30 (0.95–5.60)	0.064	3.21(0.25–10.89)	0.370
Perineural invasion (yes vs. no)	1.40 (0.59–3.50)	0.430	0.35 (0.09–1.3)	0.120
CD3 (high expression vs. low expression)	0.70 (0.25–2.00)	0.500	0.64 (0.098–4.2)	0.640
CD4 (high expression vs. low expression)	0.48 (0.21–1.10)	0.084	1.3 (0.29–6)	0.720
CD8 (high expression vs. low expression)	0.43 (0.20–0.89)	**0.023**	0.15 (0.037–0.63)	**0.010**
FoxP3 (high expression vs. low expression)	2.10 (0.92–4.80)	0.077	1.5 (0.46–4.7)	0.520
PD-L1 (high expression vs. low expression)	2.70 (1.10–7.00)	**0.036**	19 (3.5–110)	**<0.001**

*p*-values < 0.05 were considered significant and have been bolded.

## Data Availability

The datasets used for this study are available in TCGA and GEO.

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
