# Peer review of "The PD-L1 Expression and Tumor-Infiltrating Immune Cells Predict an Unfavorable Prognosis in Pancreatic Ductal Adenocarcinoma and Adenosquamous Carcinoma"

_jcm, 2023, doi:10.3390/jcm12041398_

Round 1
Reviewer 1 Report
The authors analyzed the tumor microenvironment including PD-L1 expression and tumor infiltrating immune cells in patients with pancreatic ductal adenocarcinoma and adenosquamous carcinoma.
This article had potential interests in the field of pancreatic oncology.
However, there are some concerns.
1. This study compared the PD-L1 expression and tumor infiltrating immune cells between pancreatic ductal adenocarcinoma and adenosquamous carcinoma. Thus, the title of the article must represent this point.
2. The prognosis of patients with adenosquamous carcinoma of the pancreas is worse than that of pancreatic ductal adenocarcinoma. This might be related to the tumor microenvironment. Did the results of the present study explain this prognostic difference?
3. In the present study, higher infiltration of CD3 and CD8 lymphocytes correlated with better prognosis both in pancreatic ductal adenocarcinoma and adenosquamous carcinoma, while the amounts of FoxP3-positive lymphocytes did not influence of prognosis in both carcinomas. Some papers showed that increased numbers of Fox-P3-positive lymphocytes correlated with poor prognosis. Please explain this point.
Author Response
We would like to express our sincere thanks to the reviewers for the constructive and positive comments.
Comment 1:This study compared the PD-L1 expression and tumor infiltrating immune cells between pancreatic ductal adenocarcinoma and adenosquamous carcinoma. Thus, the title of the article must represent this point.
Answer: Comment 1 advises that we should modify our title. In our previously submitted draft, we use "Tumor-Infiltrating Immune Cells Predict an Unfavorable Prognosis in Pancreatic Cancer" as the title. The title after modification is shown below: "The PD-L1 Expression and Tumor-Infiltrating Immune Cells Predict an Unfavorable Prognosis in Pancreatic Ductal Adenocarcinoma and Adenosquamous Carcinoma".
Comment 2:The prognosis of patients with adenosquamous carcinoma of the pancreas is worse than that of pancreatic ductal adenocarcinoma. This might be related to the tumor microenvironment. Did the results of the present study explain this prognostic difference?
Answer: Some studies considered that the worse prognosis of ASCP than PDAC is due to the less infiltration of immune cells. Regrettably, our study does not prove this point. This needs to be elucidated by future studies.
Comment 3:In the present study, higher infiltration of CD3 and CD8 lymphocytes correlated with better prognosis both in pancreatic ductal adenocarcinoma and adenosquamous carcinoma, while the amounts of FoxP3-positive lymphocytes did not influence of prognosis in both carcinomas. Some papers showed that increased numbers of Fox-P3-positive lymphocytes correlated with poor prognosis. Please explain this point.
Answer: During the immunohistochemical, we found the infiltration of FoxP3+ T cells is at a low level in pancreatic cancer. The difference in expression level was modest between groups with high and low FoxP3. In our study, increased numbers of FoxP3+ lymphocytes are correlated with poor prognosis. But it is not significant. More cases are needed to confirm this.
We thank the anonymous reviewers again for their constructive comments.
Reviewer 2 Report
In this manuscript, the authors aim to demonstrate the differences in immune cell composition within the TME in PDAC and ASCP, and their impact on overall survival.
Major comments:
- One of the major concerns in the paper is that the authors have highlighted how they have segregated PDAC cases from pancreatic cancer patients (parameters used, etc.) This is not mentioned in the methods section or figure 2.
- The font size in a great majority of the figures is very small and the text is not legible. Example – figures 3,5 and supplementary figures.
- Under the results section (3.3), the authors state that - “Summarily, ductal cells and epithelial cells mainly existed in the PDAC and were present at low levels in ASCP samples.” However, figure 3B tells a different story. Data from figure 3B is not supported by that in figure 3C. While there is a reduction in ductal cells, there is no reduction in epithelia (or is there a problem in color selection)? Also, when the authors state that - “The proportion of T cells and B cells was rare in ASPC and PDAC tissues, showing that TICs could be rare in PC”, is there a control cell subset from the normal pancreas that they are comparing to? Can they also show this data alongside ASCP and PDAC?
- On page 4, 4th line from the bottom, the authors state that – “Three types of TICs were negatively associated with PD-L1 expression including regulatory T cells (Tregs), activated NK cells, and activated mast cells”. If PD-L1 is negatively associated with Tregs, then how do the authors explain the immune suppressive TME? Tregs are known to play a role in tolerance. This should be discussed in the discussion section.
Minor comments:
- On page 1, line 8, the authors mentioned that – “Seurat was used to process the scRNA-seq data and CellChat was used to analysis of cell–cell communication.” Correct grammar.
- Correct for spacing on page 3, line 6.
- On page 4, under the results section, the authors state the title as – “High PD-L1 expression is associated with overall survival.” Based on the past research and data provided in the current manuscript, the authors should mention that - High PD-L1 expression is associated with shorter overall survival.
Please cite current studies from S. Leach's lab to substantiate the current findings.
Author Response
We would like to express our sincere thanks to the reviewers for the constructive and positive comments.
Major comments:
Comment 1: One of the major concerns in the paper is that the authors have highlighted how they have segregated PDAC cases from pancreatic cancer patients (parameters used, etc.) This is not mentioned in the methods section or figure 2.
Answer: The diagnostic pathology report for specimens of ASCP and PDAC was provided by the Pathology Department, West China Hospital of Sichuan University. We will add this in the methods section.
A total of 83 patients (29 ASCP patients and 54 PDAC patients) were included in the pancreatic cancer group. We will add this in figure 2.
Comment 2: The font size in a great majority of the figures is very small and the text is not legible. Example – figures 3,5 and supplementary figures.
Answer: We have enlarged the annotation marks in the revised manuscript.
Comment 3: Under the results section (3.3), the authors state that - “Summarily, ductal cells and epithelial cells mainly existed in the PDAC and were present at low levels in ASCP samples.” However, figure 3B tells a different story. Data from figure 3B is not supported by that in figure 3C. While there is a reduction in ductal cells, there is no reduction in epithelia (or is there a problem in color selection)? Also, when the authors state that - “The proportion of T cells and B cells was rare in ASPC and PDAC tissues, showing that TICs could be rare in PC”, is there a control cell subset from the normal pancreas that they are comparing to? Can they also show this data alongside ASCP and PDAC?
Answer: Thanks for your kindly review and suggestions. The color of B cells and epithelial cells is similar in figure 3B. We have changed the color of B cells in figure 3B.
According to previous study(X. Zhao, 2021), T cells and B cells were highly infiltrated (45.15%) in intraductal papillary mucinous neoplasm (IPMN). In our study, the proportion of T cells and B cells was rare in ASPC (32.98%) and PDAC (6.21%), showing that TICs could be rare in PC.
Zhao, H. Li, S. Lyu, J. Zhai, Z. Ji, Z. Zhang, X. Zhang, Z. Liu, H. Wang, J. Xu, H. Fan, J. Kou, L. Li, R. Lang, and Q. He, Single-cell transcriptomics reveals heterogeneous progression and EGFR activation in pancreatic adenosquamous carcinoma. Int J Biol Sci 17 (2021) 2590-2605.
Comment 4: On page 4, 4th line from the bottom, the authors state that – “Three types of TICs were negatively associated with PD-L1 expression including regulatory T cells (Tregs), activated NK cells, and activated mast cells”. If PD-L1 is negatively associated with Tregs, then how do the authors explain the immune suppressive TME? Tregs are known to play a role in tolerance. This should be discussed in the discussion section.
Answer: The PD-1/PD-L1 pathway can promote the naive CD4+ T cells to differentiate into regulatory T (Treg) cells. However, PD-L1 is negatively associated with Tregs in our analysis. It may be affected by extreme outliers. We have modified figure 6c.
Minor comments:
Comment 1: On page 1, line 8, the authors mentioned that – “Seurat was used to process the scRNA-seq data and CellChat was used to analysis of cell–cell communication.” Correct grammar.
Answer: The sentence was adjusted on page 1. “Seurat was used to process the scRNA-seq data and CellChat was used to analyze cell–cell communication”.
Comment 2: Correct for spacing on page 3, line 6.
Answer: We have corrected this spacing.
Comment 3: On page 4, under the results section, the authors state the title as – “High PD-L1 expression is associated with overall survival.” Based on the past research and data provided in the current manuscript, the authors should mention that - High PD-L1 expression is associated with shorter overall survival.
Answer: We have changed the title on page 4.
We have cited current studies from S. Leach's lab in the discussion part. Thank you for your valuable suggestion.
Reviewer 3 Report
The article is interesting. However, it has some limitations reducing the value of the study.
1) The sentence, especially its second part, in the second paragraph of Introduction section ("Immunotherapy drugs such as immune checkpoint inhibitors are only effective in a small percentage of PC patients, highlighting the immunosuppressive phenotype of the disease [7; 8; 9]) requires correction. It is not fully clear.
2) Figure 6D - what is source of pancreas and immune cells? Did you do them or did you use any special program being their source? Please indicate.
3) What is possible cause of detected by your research group dependence (higher levels of CD3+ and CD8+ T cell infiltration were linked with a better prognosis in PC; why not CD4+)? Please explain the possible mechanisms and causes according to among other the functions of these cells in the Discussion section.
4) There are some articles performed in the past assessing dependence between immune cells and pancreatic cancer. Are there any advantages of your work over them expect the assessment in ASCP? Please clearly indicate them in the article.
Author Response
We would like to express our sincere thanks to the reviewers for the constructive and positive comments.
Comment 1: The sentence, especially its second part, in the second paragraph of Introduction section ("Immunotherapy drugs such as immune checkpoint inhibitors are only effective in a small percentage of PC patients, highlighting the immunosuppressive phenotype of the disease [7; 8; 9]) requires correction. It is not fully clear.
Answer: The sentence after modification is shown below: “Immunotherapy drugs such as immune checkpoint inhibitors are only effective in a small percentage of PC patients, highlighting the importance of clarifying the reason for the limited efficacy of checkpoint blockade [7; 8; 9]”.
Comment 2:Figure 6D - what is source of pancreas and immune cells? Did you do them or did you use any special program being their source? Please indicate.
Answer: This figure was created by Figdraw. We will indicate this in the article.
Comment 3: What is possible cause of detected by your research group dependence (higher levels of CD3+ and CD8+ T cell infiltration were linked with a better prognosis in PC; why not CD4+)? Please explain the possible mechanisms and causes according to among other the functions of these cells in the Discussion section.
Answer: CD3 is part of the T-cell receptor/CD3 complex and is involved in T-cell development and signal transduction. The expression of CD3 is typically associated with T cells infiltrating. CD8+ cytotoxic T cells (CTL) play a vital role in recognizing and killing cancer cells. However, CD4+ cells play a complex role in tumorigenesis and tumor progression. On the one hand, CD4+Th cells and CD4+ CTL can enhance the antitumor immune response. On the other hand, CD4+ Tregs can suppress antitumor immunity. We will add this in the discussion part.
Comment 4: There are some articles performed in the past assessing dependence between immune cells and pancreatic cancer. Are there any advantages of your work over them expect the assessment in ASCP? Please clearly indicate them in the article.
Answer: Firstly, the most prominent advantage of our work is detecting immune markers in ASCP. What’s more, we chose PD-L1, which was rarely detected in previous studies, as an immune marker. Finally, we analyze the scRNA-seq data and transcriptome profiles to explain the prognostic value of TICs in pancreatic cancer. This was not seen in prior studies. We will add this in the discussion part.
We thank the anonymous reviewers again for their constructive comments.
Round 2
Reviewer 2 Report
Comment 3: Under the results section (3.3), the authors state that - “Summarily, ductal cells and epithelial cells mainly existed in the PDAC and were present at low levels in ASCP samples.” However, figure 3B tells a different story. Data from figure 3B is not supported by that in figure 3C. While there is a reduction in ductal cells, there is no reduction in epithelia (or is there a problem in color selection)? Also, when the authors state that - “The proportion of T cells and B cells was rare in ASPC and PDAC tissues, showing that TICs could be rare in PC”, is there a control cell subset from the normal pancreas that they are comparing to? Can they also show this data alongside ASCP and PDAC?
Answer: Thanks for your kindly review and suggestions. The color of B cells and epithelial cells is similar in figure 3B. We have changed the color of B cells in figure 3B.
According to previous study(X. Zhao, 2021), T cells and B cells were highly infiltrated (45.15%) in intraductal papillary mucinous neoplasm (IPMN). In our study, the proportion of T cells and B cells was rare in ASPC (32.98%) and PDAC (6.21%), showing that TICs could be rare in PC.
Zhao, H. Li, S. Lyu, J. Zhai, Z. Ji, Z. Zhang, X. Zhang, Z. Liu, H. Wang, J. Xu, H. Fan, J. Kou, L. Li, R. Lang, and Q. He, Single-cell transcriptomics reveals heterogeneous progression and EGFR activation in pancreatic adenosquamous carcinoma. Int J Biol Sci 17 (2021) 2590-2605.
Query: From figures 3B and 3C, it is clear that T- and B-cells are present more so in ASPC than in PDAC. The argument made above by the authors is a little difficult to understand. Please clarify. Please give separate % values for each T- and B-cells in PDAC and ASPC, respectively.
Comment 4: On page 4, 4th line from the bottom, the authors state that – “Three types of TICs were negatively associated with PD-L1 expression including regulatory T cells (Tregs), activated NK cells, and activated mast cells”. If PD-L1 is negatively associated with Tregs, then how do the authors explain the immune suppressive TME? Tregs are known to play a role in tolerance. This should be discussed in the discussion section.
Answer: The PD-1/PD-L1 pathway can promote the naive CD4+ T cells to differentiate into regulatory T (Treg) cells. However, PD-L1 is negatively associated with Tregs in our analysis. It may be affected by extreme outliers. We have modified figure 6c.
Query: The authors state that the correlation may be affected by extreme outliers, and in doing so they completely removed the figure panel for Treg cells from figure 6C. They further state on page 5 that – “Two types of TICs were negatively associated with PD-L1 expression including 233 activated NK cells and activated mast cells (Fig. 6B-C).”
This is not justified. The negative correlation R-value for NK cells is -0.23 and for Treg is -0.29. The authors can not just remove data in this manner. Please provide valid reasons/discuss in the discussion section.
Author Response
Comment 3: Under the results section (3.3), the authors state that - “Summarily, ductal cells and epithelial cells mainly existed in the PDAC and were present at low levels in ASCP samples.” However, figure 3B tells a different story. Data from figure 3B is not supported by that in figure 3C. While there is a reduction in ductal cells, there is no reduction in epithelia (or is there a problem in color selection)? Also, when the authors state that - “The proportion of T cells and B cells was rare in ASPC and PDAC tissues, showing that TICs could be rare in PC”, is there a control cell subset from the normal pancreas that they are comparing to? Can they also show this data alongside ASCP and PDAC?
Answer: Thanks for your kindly review and suggestions. The color of B cells and epithelial cells is similar in figure 3B. We have changed the color of B cells in figure 3B.
According to previous study(X. Zhao, 2021), T cells and B cells were highly infiltrated (45.15%) in intraductal papillary mucinous neoplasm (IPMN). In our study, the proportion of T cells and B cells was rare in ASPC (32.98%) and PDAC (6.21%), showing that TICs could be rare in PC.
Zhao, H. Li, S. Lyu, J. Zhai, Z. Ji, Z. Zhang, X. Zhang, Z. Liu, H. Wang, J. Xu, H. Fan, J. Kou, L. Li, R. Lang, and Q. He, Single-cell transcriptomics reveals heterogeneous progression and EGFR activation in pancreatic adenosquamous carcinoma. Int J Biol Sci 17 (2021) 2590-2605.
Query: From figures 3B and 3C, it is clear that T- and B-cells are present more so in ASPC than in PDAC. The argument made above by the authors is a little difficult to understand. Please clarify. Please give separate % values for each T- and B-cells in PDAC and ASPC, respectively.
Answer: Once again, thank you very much for your comments and suggestions. In the previous study(X. Zhao, 2021), the researcher compares the T cells and B cells infiltration among intraductal papillary mucinous neoplasm (IPMN), PDAC, and ASPC. We have modified this part now.
Page 5: In our study, the proportion of T cells(3.50%) and B cells(2.71%) was rare in PDAC. However, T cells(9.26%) and B cells(23.73%) were highly infiltrated in ASPC.
Page 15: Some studies considered that the worse prognosis of ASCP than PDAC is due to the less infiltration of immune cells. Regrettably, our study does not prove this point, which needs to be elucidated by future studies.
Comment 4: On page 4, 4th line from the bottom, the authors state that – “Three types of TICs were negatively associated with PD-L1 expression including regulatory T cells (Tregs), activated NK cells, and activated mast cells”. If PD-L1 is negatively associated with Tregs, then how do the authors explain the immune suppressive TME? Tregs are known to play a role in tolerance. This should be discussed in the discussion section.
Answer: The PD-1/PD-L1 pathway can promote the naive CD4+ T cells to differentiate into regulatory T (Treg) cells. However, PD-L1 is negatively associated with Tregs in our analysis. It may be affected by extreme outliers. We have modified figure 6c.
Query: The authors state that the correlation may be affected by extreme outliers, and in doing so they completely removed the figure panel for Treg cells from figure 6C. They further state on page 5 that – “Two types of TICs were negatively associated with PD-L1 expression including 233 activated NK cells and activated mast cells (Fig. 6B-C).”
This is not justified. The negative correlation R-value for NK cells is -0.23 and for Treg is -0.29. The authors can not just remove data in this manner. Please provide valid reasons/discuss in the discussion section.
Answer: Thank you for your suggestion. As suggested by reviewer, we have added the suggested content to the manuscript in the discussion section and recovered figure 6c.
The PD1/PD-L1 pathway and Treg cells are both known to contribute to tumor immune tolerance. Remarkably, we found high expression of PD-L1 was associated with low infiltration of Treg cells. We considered that the PD-1/PD-L1 axis could inhibit Ras/MEK/ERK pathway to repress T cell proliferation, including Treg cells. In addition, the PD-1/PD-L1 axis leads to a decrease in cell apoptosis-related gene Bcl-xl and promotes Treg cell apoptosis. The high level of PD-1/PD-L1 expression resulted in the apoptosis of Treg cells and the decrease of proliferation, leading to the reduction of Treg cells in the TME.
Reviewer 3 Report
Authors regard to all suggestions and questions, and changed indicated issues.
Author Response
Once again, thank you very much for your comments and suggestions.